# Molecular Regulation of Photosynthetic Carbon Assimilation in Oat Leaves Under Drought Stress

**DOI:** 10.3390/plants13233317

**Published:** 2024-11-26

**Authors:** Yiqun Xu, Liling Jiang, Jia Gao, Wei Zhang, Meijun Zhang, Changlai Liu, Juqing Jia

**Affiliations:** 1College of Agriculture, Shanxi Agricultural University, Jinzhong 030810, China; 2Academy of Agricultural and Forestry Sciences, Qinghai University National Duplicate Genebank for Crops, Xining 810016, China; 3Co-Innovation Center for Sustainable Forestry in Southern China, Bamboo Research Institute, Nanjing Forestry University, Nanjing 210037, China; 4Houji Laboratory in Shanxi Province, Academy of Agronomy, Shanxi Agricultural University, Taiyuan 030031, China

**Keywords:** common oat, PEG stress, transcriptome analysis, differentially expressed genes

## Abstract

Common oat (*Avena sativa* L.) is one of the important minor grain crops in China, and drought stress severely affects its yield and quality. To investigate the drought resistance characteristics of oat seedlings, this study used Baiyan 2, an oat cultivar at the three-leaf stage, as the experimental material. Drought stress was simulated using polyethylene glycol (PEG) to treat the seedlings. The photosynthetic parameters and physicochemical indices of the treatment groups at 6 h and 12 h were measured and compared with the control group at 0 h. The results showed that drought stress did not significantly change chlorophyll content, but it significantly reduced net photosynthetic rate and other photosynthetic parameters while significantly increasing proline content. Transcriptome analysis was conducted using seedlings from both the control and treatment groups, comparing the two treatment groups with the control group using Tbtool software (v2.136). This analysis identified 344 differentially expressed genes. Enrichment analysis of these differentially expressed genes revealed significant enrichment in physiological pathways such as photosynthesis and ion transport. Ten differentially expressed genes related to the physiological process of photosynthetic carbon assimilation were identified, all of which were downregulated. Additionally, seven differentially expressed genes were related to ion transport. Through gene co-expression analysis combined with promoter region structure analysis, 11 transcription factors (from *MYB*, *AP2/ERF*, *C2C2-dof*) were found to regulate the expression of 10 genes related to photosynthetic carbon assimilation. Additionally, five transcription factors regulate the expression of two malate transporter protein-related genes (from *LOB*, *zf-HD*, *C2C2-Dof*, etc.), five transcription factors regulate the expression of two metal ion transporter protein-related genes (from *MYB*, *zf-HD*, *C2C2-Dof*), five transcription factors regulate the expression of two chloride channel protein-related genes (from *MYB*, *bZIP*, *AP2/ERF*), and two transcription factors regulate the expression of one Annexin-related gene (from *NAC*, *MYB*). This study provides a theoretical foundation for further research on the molecular regulation of guard cells and offers a molecular basis for enhancing drought resistance in oats.

## 1. Introduction

Common oat (*Avena sativa* L.) is an annual coarse grain crop. Due to its rich nutritional value, it has garnered increasing attention worldwide [1]. In China, the main cultivation areas for oats are located in the arid and semi-arid regions of Shanxi, Shaanxi, Qinghai, and Gansu. Consequently, drought is one of the major factors affecting oat yield [2]. Plants responding to drought stress involve a complex reaction mechanism that includes multiple genes, signal pathways, and metabolic processes [3]. Research indicates that severe drought stress can lead to a decrease in oat fertility, reduced plant height [4], fewer panicles, and yellowing or chlorosis of leaves [5], which in turn causes a significant reduction in oat yield [6]. Therefore, identifying drought-resistant genes in oats is crucial for oat genetic breeding.

To cope with drought stress, plants have evolved numerous mechanisms over a long period, such as enhancing their water storage capacity, producing compounds like proline to maintain osmotic pressure within cells [7], and regulating the opening and closing of stomata on leaves to reduce transpiration and minimize water loss [8,9]. Stomata are the primary organs for transpiration in plants, and their opening and closing are controlled by guard cells [10]. When guard cells absorb water and swell, stomata open; conversely, they close [11]. Current research has found that the water absorption and loss in guard cells are regulated by the osmotic pressure within the cells, which is primarily maintained by malic acid, sucrose, K ions, and Cl ions within the guard cells [12,13]. The concentration of these substances can be regulated through ion channels on the plasma membrane and vacuolar membrane [14], and the activity of these ion channels is influenced by various factors, among which calcium signaling is the primary pathway regulating guard cells [15]. It is currently believed that drought stress leads to an increase in cytoplasmic Ca^2+^ concentration in guard cells, which can promote the binding of calcineurin B-like proteins (CBLs) with protein kinase CIPK, forming a Ca^2+^-CBL-CIPK complex [16], activating anion channel proteins on the plasma membrane, and increasing anion efflux while simultaneously inhibiting K^+^ channels on the plasma membrane, reducing K^+^ influx into guard cells [17].

Stomata are essential channels for CO_2_ to enter the leaf. When plants experience drought stress, stomatal limitation significantly reduces photosynthesis [18]. This is because stomatal closure leads to lower stomatal conductance, reducing CO_2_ absorption. Insufficient CO_2_, the photosynthetic substrate, decreases the activity of key photosynthetic carbon assimilation enzymes, such as Rubisco [19]. Studies have shown that stress conditions inhibit the expression of genes related to photosynthetic carbon assimilation enzymes, leading to reduced activities of enzymes like ribulose-1,5-bisphosphate carboxylase/oxygenase (Rubisco) and fructose-1,6-bisphosphate aldolase (FBA), thus hindering the photosynthetic process [20,21,22]. Research indicates that drought-tolerant plants maintain relatively high activity and transcription levels of Rubisco under drought stress [23]. Yamada et al. [24] found that overexpression of FBA can increase proline content in transgenic plants under salt stress, thereby enhancing stress resistance. Zeng et al. [25] discovered that overexpression of the FBA gene can promote plant growth under adverse conditions. However, the dynamic changes in photosynthetic parameters and physicochemical indicators related to these physiological activities in oats responding to drought stress, as well as the regulation of these genes, remain largely unknown.

When plants encounter environmental stress, transcription factors can regulate the expression of multiple stress-related genes to enhance plant stress resistance. Numerous transcription factors involved in stress response have been isolated. Based on current research findings, the transcription factors involved in plant stress responses are primarily of the following types: *AP2/EREBP*, *bZip*, *MYB*, and others. For example, overexpression of the ABF2 transcription factor from the bZip family can enhance drought resistance in transgenic crops [26]. The *OsNAC033* gene in rice can confer greater drought and salt tolerance to transgenic plants [27]. The transcription factor *CBF3* from the *AP2/EREBP* family can significantly increase the frost resistance of plants [28].

This study used the drought-tolerant oat (*Avena sativa* L.) cultivar Baiyan 2 as the experimental material. By sampling oat seedlings subjected to PEG-simulated drought stress at different times, photosynthetic parameters and physicochemical indicators were measured. Based on transcriptome data analysis of oats under drought induction, the signaling pathways of oat response to drought stress were analyzed, and relevant regulatory genes were identified, providing a foundation for research on the drought resistance mechanisms of oat.

## 2. Materials and Methods

### 2.1. Materials

Baiyan 2, a highly drought-tolerant common oat (*Avena sativa* L.) cultivar, was used as experimental material; the seeds were soaked in 75% alcohol for 1 min, then soaked in 15% sodium hypochlorite for 20 min, and finally rinsed 5 times with sterile distilled water and air-dried for later use.

### 2.2. Methods

#### 2.2.1. PEG Stress Concentration Screening

Using PEG-6000 (Merck Millopore, Burlington, MA, USA) as the drought stress treatment reagent, four concentrations of PEG-6000 were set up for treatment: 0% (control group), 10%, 20%, and 30%. The disinfected, cleaned, and air-dried seeds were evenly placed in disposable petri dishes with double-layer filter paper, and 100 seeds per dish. With one petri dish for one repetition, each treatment was repeated three times. An 8 mL stress solution of the corresponding concentration was added to each treated petri dish. For each treatment, seven seedlings were randomly selected to measure the length of the primary root and the plant height after 9 d drought treatment.

#### 2.2.2. Drought Stress

The disinfected oat seeds were placed in a seedling tray for germination. After 4 days, seeds with consistent growth were transferred to a hydroponic box with a 96-well plate and Hoagland’s nutrient solution was added. The seedlings were placed in a light incubator for normal growth for 10 days, with the nutrient solution changed every other day. The conditions of the light incubator were set to a constant temperature of 26 °C, a relative humidity of 50–55%, and 12 h of light per day (10,000 Lux). A 20% PEG6000 + Hoagland nutrient solution was prepared. On the 11th day, the Hoagland nutrient solution in the hydroponic box was replaced with a 20% PEG6000 + Hoagland nutrient solution to simulate PEG stress. Complete seedlings were collected at 0 h (CK), 6 h (mild drought stress), and 12 h (severe drought stress), with each treatment repeated three times biologically. The samples were frozen in liquid nitrogen and stored at −80 °C for RNA extraction.

#### 2.2.3. Measurement of Physicochemical Indicator Content

Chlorophyll was extracted using the ethanol extraction method [29]. Select 10-day-old oat seedlings treated with 20% PEG-6000 stress for 0 h, 6 h, and 12 h. After grinding, weigh 0.5 g of powder and add it to a 1:35 acetone: ethanol extraction solution (V:V = 1:1). Soak for 24 h under dark room temperature conditions. Use a spectrophotometer to measure the absorbance of the extraction solution at 470 nm, 645 nm, 653 nm, and 663 nm. Calculate the chlorophyll content according to the following formula:Chlorophyll a content: Chl a (µg·g^−1^) = (12.71D_663_−2.59D_653_) × V_e_/M_f_;
Chlorophyll b content: Chl b (µg·g^−1^) = (20.13D_645_−4.67D_663_) × V_e_/M_f_;
Total Chlorophyll Content = Chl a + Chl b.

Proline content was measured according to Bates et al. [30]. Use proline standard to establish a standard curve. Weigh 0.5 g of oat powder sample, homogenize it in 5 mL of 3% sulfosalicylic acid aqueous solution, and boil for 10 min. After cooling, filter into a clean test tube, and this filtrate is the proline extract. Take 2 mL of the filtrate and add 2 mL of glacial acetic acid and 2 mL of acidic ninhydrin reagent, heat in a 100 °C boiling water bath for 30 min, and the solution turns red. After cooling, add 4 mL of toluene, shake for 30 s, then centrifuge at 3000 rpm for 5 min. Take the upper layer of red toluene solution of proline, use toluene as a blank control, read the absorbance value at 520 nm on the spectrophotometer, and calculate the proline content according to the standard curve.

#### 2.2.4. Measurement of Photosynthetic Index

The measurement of photosynthetic parameters was conducted following the method described by Chen et al. [31]. The photosynthetic index of the leaves was measured using a LI-6400XT (LI-COR, Lincoln, NE, USA) portable photosynthesis meter. The light intensity inside the leaf chamber was 1200 μmol·m^−2^·s^−1^, and the CO_2_ concentration was 400 μmol·mol^−1^. Ten-day-old oat seedlings growing normally were set as the control group, and ten-day-old oat seedlings treated with 20% PEG-6000 were set as the treatment group. At 6 h and 12 h after PEG treatment, the middle part of the fully expanded third leaf was selected for measurement to obtain the net photosynthetic rate, transpiration rate, and stomatal conductance. Each leaf was measured twice technically and the average value was taken.

#### 2.2.5. RNA Extraction, Library Preparation, and Sequencing

After wrapping the three sets of samples in dry ice and shipping them to Novogene Co., Ltd. in Beijing, China, total RNA from the three sets of samples was extracted using the TRIzol method. The RNA samples were quality controlled using an Agilent 2100 bioanalyzer (Agilent, Santa Clara, CA, USA). After the RNA samples passed the quality check, a library was constructed using the NEB standard library construction method. After the library was constructed, preliminary quantification was performed using a Qubit2.0 Fluorometer (Thermo Fisher Scientific, Waltham, MA, USA), and the library was diluted to 1.5 ng/μL. The insert size of the library was then checked using an Agilent 2100 bioanalyzer (Agilent, Santa Clara, CA, USA). After the insert size met expectations, the effective concentration of the library was accurately quantified using qRT-PCR (the effective concentration of the library was higher than 1.5 nM) to ensure the quality of the library. After the library passed the quality check, different libraries were pooled according to the effective concentration and the required amount of data for downstream sequencing and then sequenced using Illumina sequencing [32].

#### 2.2.6. Quality Control of Sequencing Data

Raw data in fastq format was obtained through sequencing. To ensure the quality and reliability of data analysis, the software fastp (version 0.19.7) was used for data quality control, filtering out reads with adapters, reads containing N (N indicates that base information cannot be determined), and low-quality reads (reads where the number of bases with Qphred ≤ 5 accounts for more than 50% of the entire read length) from the raw data. We obtained 25.2G of clean data. At the same time, Q20 and Q30 calculations were performed on the clean data.

#### 2.2.7. Reference Genome Alignment

The HISAT2 software (2.0.5) was used to align the quality-controlled clean reads with the ‘sanfensan’ reference genome, obtaining the location information of the Reads on the reference genome [33].

#### 2.2.8. Differential Gene Screening

Gene expression values in RNA-seq are generally represented by FPKM. Genes with significant differences in expression levels under different conditions are screened based on the criteria of |log2(FoldChange)| ≥ 1 and padj ≤ 0.05 [34].

#### 2.2.9. GO and KEGG Enrichment Analysis of Differentially Expressed Genes

After obtaining differentially expressed genes, GO and KEGG enrichment analyses are performed on the selected genes. By analyzing the KEGG and GO enrichment results, combined with physicochemical data, key genes are identified [35].

#### 2.2.10. Gene Correlation Analysis

Using R 4.4.1 (https://www.r-project.org/), perform gene expression data correlation analysis using the criteria (r > 0.8 or r < −0.8 and *p* < 0.05) as the screening standard. Cytoscape 3.10.2 software will be used for visualization [36].

#### 2.2.11. Promoter Binding Site Analysis

Use TBtools (v2.136) [37] ‘Gtf/Gff3 Sequence Extractor’ to extract promoter sequences of key genes, and then input the obtained promoter sequences into FIMO (https://meme-suite.org/meme/tools/fimo) (accessed on 20 October 2024) [38] to predict the binding sites of key gene promoter sequences.

#### 2.2.12. Statistical Analysis

The data was organized using Microsoft Excel 2019, and a two-way ANOVA test was conducted using GraphPad Prism 9 software with a significance level of *p* = 0.05. Relevant charts were also created using this software.

## 3. Results

### 3.1. Selection of PEG Treatment Concentration

To screen for the most significant stress effect of PEG-6000 concentration, the height and root length of oat seedlings were evaluated at different concentrations. As shown in Figure 1A, after treating exposed oat seeds with PEG-6000 for 9 days, the oat seeds still exhibited only seed exposure without root or shoot growth under a 30% PEG treatment. With a 20% PEG treatment, a small number of shoots and roots emerged. In the 10% PEG treatment, obvious roots and shoots grew, and the shoots had already formed stems. However, compared to the control group without PEG stress treatment, there was still a growth inhibition phenomenon of roots and stems. Combined with specific measurements of root length and plant height (Figure 1B,C), it was found that 30% PEG severely inhibited the growth of roots and shoots, while 20% had a significant inhibitory effect on the elongation of roots and stems. Therefore, for the subsequent drought simulation treatment of oat seedlings in this study, 20% PEG-6000 was used.

### 3.2. Effects of Drought Treatment on Physicochemical Indicators

To study the dynamic changes in oats under 20% PEG stress treatment, chlorophyll and proline contents in leaves were measured at three time points (0 h, 6 h, 12 h) after drought treatment. These two physicochemical indicators play an important role in maintaining normal physiological functions of plants under drought stress. As shown in Figure 2, the total chlorophyll content (Chl a + b) in the 6 h and 12 h treatment groups did not show significant changes compared to the 0 h control group during short-term drought stress (Figure 2A). However, the proline content increased significantly (Figure 2B), which indicates that 12 h of drought stress does not affect the chlorophyll content in oat leaves but induces an increase in proline content, generating a response to drought stress.

### 3.3. Effects of PEG-6000 Treatment on Photosynthetic Parameters in Oats

By comparing the control group (0 h, control group) and the treatment group (6 h, 12 h drought), it was observed that the net photosynthetic rate of the treatment group (drought) significantly decreased at 6 h compared to the control group (CK) (Figure 3A). The stomatal conductance, transpiration rate, and intercellular CO_2_ concentration of the treatment group also decreased as the duration of stress increased (Figure 3B–D).

### 3.4. Quality Control Analysis of Transcriptome Sequencing

By collecting oat seedling samples under 20% PEG stress at 0 h, 6 h, and 12 h and extracting their total RNA, nine libraries were constructed. The transcriptome data results are shown in Appendix A. Based on the Illumina sequencing platform, 65.95G of raw data were obtained from the nine sample libraries (baiyan2_0_1, baiyan2_0_2, baiyan2_0_3, baiyan2_6_1, baiyan2_6_2, baiyan2_6_3, baiyan2_12_1, baiyan2_12_2, baiyan2_12_3), and after filtering, 62.19G of clean bases were acquired. The base error rate was 0.01%, Q20 ranged from 97.61 to 98.57, and Q30 was above 93%. The GC content was between 53.64% and 55.85%. These results indicate that the overall sequencing quality of the three groups of samples was good, and the sequencing reads were successfully aligned to the genome at a very high rate, supporting subsequent analyses. Pearson correlation analysis showed that the biological replicates of the three groups were highly reproducible (Figure 4). The results confirm that the RNA data of baiyan2_0, baiyan2_6, and baiyan2_12 are reliable.

### 3.5. Identification of Differentially Expressed Genes Under Drought Stress

Using the drought-treated 0 h group as the control group, and the drought-treated 6 h (mild drought stress) and 12 h (severe drought stress) groups as the experimental groups, genes that met the criteria of *p*-adjusted value ≤ 0.05 and |log2FoldChange| ≥ 1.0 were selected as significantly differentially expressed genes (DEGs). Based on the significance results, compared to the 0 h control group, 2919 DEGs were obtained at 6 h (mild drought stress), with 1411 DEGs upregulated and 1508 DEGs downregulated. At 12 h (severe drought stress), a total of 8429 DEGs were obtained, with 3939 upregulated and 4490 downregulated. Additionally, 1012 DEGs were induced only at 6 h (mild drought stress), 3143 DEGs were induced only at 12 h (severe drought stress), and 1460 DEGs were induced by drought stress at both time points. There were 344 genes consistently expressed across the comparisons baiyan2_12 vs. baiyan2_6, baiyan2_12 vs. baiyan2_0, and baiyan2_6 vs. baiyan2_0 (Figure 5). Based on the expression patterns and clustering heatmap of the 344 DEGs, these 344 genes can be divided into three groups: down group (downregulated at both 6 h and 12 h vs. 0 h), up group (upregulated at 12 h), and middle group (upregulated at 6 h and downregulated at 12 h) (Figure 6). The down group has 242 genes, the up group has 73 genes, and the middle group has 29 genes (Figure 6).

### 3.6. GO Enrichment Analysis of Differentially Expressed Genes

By performing GO enrichment analysis on the significantly differentially expressed genes, these genes can be categorized into three groups: Biological Process (BP), Cellular Component (CC), and Molecular Function (MF) (Figure 7). The 344 DEGs were enriched into a total of 49 subcategories. Among these, the Biological Process (BP) category includes 33 subcategories, the Cellular Component (CC) category includes 3 subcategories, and the Molecular Function (MF) category includes 13 subcategories. In the Biological Process (BP) category, 344 differentially expressed genes were primarily enriched in carbohydrate metabolism (e.g., carbohydrate catabolic process), amino acid metabolism (e.g., serine family amino acid metabolic process), nucleotide compound metabolism (e.g., nucleotide catabolic process), photosynthesis, and ion transport (e.g., metal ion transport, chloride transport). In the Cellular Component (CC) category, differentially expressed genes were primarily enriched in components of the photosystem. In the Molecular Function (MF) category, differentially expressed genes were primarily enriched in subcategories such as fructose-bisphosphate aldolase activity, glucose-6-phosphate dehydrogenase activity, gated channel activity, and metal cluster binding.

GO enrichment analysis of differentially expressed genes with distinct expression characteristics revealed that the up group of differentially expressed genes were primarily enriched in subcategories such as chloride transport and voltage-gated monoatomic anion channel activity, whereas the down group of differentially expressed genes were mainly enriched in molecular function categories including fructose-bisphosphate aldolase activity and aldehyde-lyase activity, as well as in Biological Process (BP) categories such as organic acid catabolic process, glucose metabolic process, and photosynthesis. The differentially expressed genes in the middle group were primarily enriched in molecular function categories such as intramolecular phosphotransferase activity and in Biological Process (BP) categories such as pyrimidine-containing compound biosynthetic process.

### 3.7. KEGG Enrichment Analysis of Differentially Expressed Genes

To further understand the metabolic processes involved by differentially expressed genes (DEGs), KEGG enrichment analysis was conducted on 344 DEGs, as well as DEGs in the up group and down group, with a threshold of *p* value < 0.05. The results showed that DEGs were mainly enriched in metabolic processes such as photosynthesis—antenna proteins, porphyrin metabolism, etc (Figure 8A). Seven genes were enriched in carbon fixation in photosynthetic organisms, and five DEGs were enriched in the pentose phosphate pathway. Other DEGs were primarily enriched in metabolic processes related to the biosynthesis of amino acids such as glycine, serine, and threonine, ascorbate, cyanoamino acid, and lysine, as well as the biosynthesis of substances like monobactams and carotenoids (Figure 8A). Further analysis of the down group DEGs revealed that they were mainly enriched in metabolic processes such as photosynthesis—antenna proteins, porphyrin metabolism, and carbon fixation in photosynthetic organisms. Basically, the metabolic processes enriched with 344 differentially expressed genes are consistent, except for the lack of enrichment of differentially expressed genes in ascorbate and aldarate metabolism (Figure 8B). In the up group, differentially expressed genes are enriched in glycerolipid, ascorbate, and aldarate, as well as histidine, beta-alanine, arginine, and proline amino acid metabolism. Analysis of gene enrichment in the up group shows that two genes are enriched in the proline metabolic pathway, consistent with the previous results of proline content (Figure 8C). Additionally, three differentially expressed genes in the up group are significantly enriched in ascorbate and aldarate metabolism, indicating that although the ascorbic acid content in seedlings was not measured in this study, transcriptome data analysis suggests that drought stress should lead to an increase in ascorbic acid content in seedlings.

### 3.8. Selection of Key Differentially Expressed Genes

To gain a more detailed understanding of drought resistance in oats, this study selected genes involved in photosynthetic carbon assimilation and carbohydrate metabolism, ion channel protein activity, and calcium signaling as key genes. The physiological processes included in photosynthetic carbon assimilation and carbohydrate metabolism are: carbon fixation in photosynthetic organisms, the pyruvate biosynthetic process, pyruvate metabolic process, carbohydrate catabolic process, generation of precursor metabolites and energy, monocarboxylic acid metabolic process, fructose-bisphosphate aldolase activity, pentose phosphate pathway, glucose-6-phosphate dehydrogenase activity, hexose metabolic process, and monosaccharide metabolic process (Figure 9A). A total of 11 differentially expressed genes were identified. Six proteins were encoded, with specific functions detailed in Table 1. Among them, P00871, Q40004, P12782, Q40677, and Q8LK61 are enzymes involved in the light-dependent reactions of photosynthesis and Calvin cycle, while Q43839 is a key enzyme in the pentose phosphate pathway. The physiological processes associated with ion channel protein activity include sodium ion transport, metal ion transport, chloride transport, inorganic anion transmembrane transporter activity, anion transmembrane transporter activity, and gated channel activity (Figure 9B). A total of six differentially expressed genes encode four proteins, with Q8L7Z9 and Q8LG88 encoding malate transporters, Q9C5D3 being a metal ion channel protein, Q96282 a chloride channel protein, and Q94CK4 a calcium-dependent phospholipid binding protein. Analysis of the physiological processes and gene expression chord diagram (Figure 9) shows that all five proteins involved in photosynthetic carbon assimilation, as well as the gene for protein Q43839, exhibit downregulated expression, indicating that drought stress reduces photosynthetic carbon assimilation and the pentose phosphate pathway in oats. The malate transporter and metal ion channel protein genes showed downregulated expression, while the chloride ion transporter and calcium-dependent phospholipid binding protein-encoding genes exhibited upregulated expression. This indicates that under drought stress, the capacity for metal ion and malate transport decreases, while the capacity for chloride and calcium ion transport increases.

### 3.9. Gene Co-Expression Analysis

Transcription factor annotation analysis was performed on 344 differentially expressed genes, resulting in the identification of 53 transcription factor genes. To obtain the transcription factors regulating 17 key genes, co-expression analysis was conducted between the 53 transcription factor genes and the 17 functional genes, as shown in Figure 10. It was found that 34 transcription factors, including *A_satnudSFS7D01G000031* and *A_satnudSFS7A01G001127*, exhibited strong correlations (*r* > 0.8 or *r* < −0.8 and *p* < 0.05) with the selected 10 key genes involved in photosynthetic carbon assimilation and carbohydrate metabolism. Additionally, all 53 transcription factors showed strong correlations (*r* > 0.8 or *r* < −0.8 and *p* < 0.05) with the seven key genes involved in ion channel activity.

Base gene co-expression analysis (Figure 10) revealed that 22 transcription factors may be involved in regulating the synthesis of Rubisco, 32 transcription factors may be involved in regulating the synthesis of Glucose-6-phosphate 1-dehydrogenase, 22 transcription factors may be involved in regulating the synthesis of phosphoglycerate kinase, 34 transcription factors may be involved in regulating fructose-bisphosphate aldolase, and 24 transcription factors may be involved in regulating NADP-dependent glyceraldehyde-3-phosphate dehydrogenase. And there are 32 transcription factors that may influence the regulation of differentially expressed genes for dicarboxylate transport proteins. Additionally, 41 transcription factors show high correlation with the regulation of differentially expressed genes for heavy-metal-associated isoprenylated plant proteins, 9 transcription factors are highly correlated with the regulation of the Annexin gene *A_satnudSFS5C01G003567*, and 13 transcription factors are highly correlated with chloride ion-channel protein genes.

### 3.10. Transcription Factor Binding Analysis

To more accurately determine the regulatory effects of transcription factors on key genes, promoter sequences of these key genes were obtained based on their chromosomal location information. The promoter sequences were analyzed for transcription factor binding sites. Integrating the analysis of transcription factor binding sites with gene co-expression analysis, it was found that 11 transcription factors have binding sites in the promoter regions of 10 key genes related to photosynthetic carbon assimilation and carbohydrate metabolism, and 15 transcription factors that have corresponding binding sites within 7 key genes related to ion channel (Figure 11 and Table 2).

In the promoters of Rubisco regulatory genes *A_satnudSFS6A01G005281* and *A_satnudSFS6A01G005280*, there are binding sites for two types of transcription factors: *CDF2* and *CDF3*. Consequently, they can bind to *A_satnudSFS4C01G000546*, *A_satnudSFS1D01G000953* and *A_satnudSFS4A01G004380*.

The promoters of phosphoglycerate kinase regulatory genes *A_satnudSFS1A01G004166* and *A_satnudSFS1D01G003897* contain binding sites for the following three transcription factors: *A_satnudSFS4C01G000546*, *A_satnudSFS1D01G000953*, and *A_satnudSFS4A01G004380*. Additionally, another regulatory gene, *A_satnudSFS1A01G001130*, has binding sites for the following eight transcription factors: *A_satnudSFS7A01G001127*, *A_satnudSFS6A01G001999*, *A_satnudSFS6C01G005376*, *A_satnudSFS7D01G002429*, *A_satnudSFS4C01G000546*, *A_satnudSFS5C01G004248*, *A_satnudSFS1D01G000953*, *and A_satnudSFS4A01G004380*.

In the promoters of glucose-6-phosphate 1-dehydrogenase regulatory genes *A_satnudSFS4A01G001930* and *A_satnudSFS4D01G005410*, binding sites for *LHY* and *RVE6*-type transcription factors were identified. Additionally, in the promoter of *A_satnudSFS4D01G005410*, binding sites for *ZHD5* type transcription factors *A_satnudSFS3D01G001960* and *A_satnudSFS3A01G002945*, as well as *DREB1A*-type transcription factor *A_satnudSFS5C01G004248*, were found.

In the promoters of fructose-bisphosphate aldolase regulatory genes *A_satnudSFS4D01G005932* and *A_satnudSFS4A01G001345*, the following five transcription factor binding sites were identified: *A_satnudSFS7A01G001127*, *A_satnudSFS6A01G001999*, *A_satnudSFS6C01G005376*, *A_satnudSFS7D01G002429*, and *A_satnudSFS4C01G000546*.

In the promoter of the NADP-dependent glyceraldehyde-3-phosphate dehydrogenase regulatory gene *A_satnudSFS5C01G000964*, binding sites for *RVE6*, *LHY*, *CDF2*, and *CDF3*-type transcription factors were identified.

There are 15 transcription factors that have corresponding binding sites within key ion channel genes.

In the regulatory gene *A_satnudSFS5C01G003567* for the Annexin, binding sites for *A_satnudSFS2D01G002820* and *A_satnudSFS2C01G003971* were identified.

In the regulatory gene *A_satnudSFS3C01G002643* for dicarboxylate transport proteins, binding sites for transcription factors such as *A_satnudSFS4D01G001891*, *A_satnudSFS4D01G005468*, and *A_satnudSFS3D01G001960* were identified. Additionally, in another key gene involved in the regulation of dicarboxylate transport proteins, *A_satnudSFS5C01G000761*, binding sites for *A_satnudSFS4C01G000546* and *A_satnudSFS5C01G004248* were found in its promoter.

In the promoter of the regulatory gene *A_satnudSFS2C01G002516*, which is related to heavy-metal-associated isoprenylated plant proteins, the following five transcription factor binding sites were found: *A_satnudSFS6A01G001999*, *A_satnudSFS6C01G005376*, *A_satnudSFS3D01G001960*, *A_satnudSFS3A01G002945*, and *A_satnudSFS4C01G000546*. In contrast, in the promoter of another regulatory gene, *A_satnudSFS6A01G001672*, only the binding site for the transcription factor *A_satnudSFS4C01G000546* was found.

In the promoter of the chloride channel protein regulatory gene *A_satnudSFS2C01G002908*, binding sites for the following three transcription factors were identified: *A_satnudSFS2C01G003971*, *A_satnudSFS6D01G001494*, and *A_satnudSFS7C01G002678*. In another regulatory gene, *A_satnudSFS2D01G006819*, binding sites for transcription factors *A_satnudSFS2D01G000591* and *A_satnudSFS7A01G002691* were identified. Both of these transcription factors belong to the *ERF2* type.

Types of transcription factors with binding sites can be found in Table 2.

## 4. Discussion

Drought is one of the primary abiotic stresses affecting crop growth, development, and quality formation [2]. The mechanism of plant response to drought stress involves multiple aspects and layers of molecular mechanisms [3]. Studies have found that under drought stress conditions, oat seedlings significantly upregulate the levels of various small molecules, such as proline, which play a crucial role in osmotic regulation. This helps maintain cell osmotic pressure, reduce stomatal conductance, and mitigate the potential harmful effects of drought stress on normal plant metabolism [39].

By measuring various physiological indicators of the control group and treatment group, it was found that, compared to the control group, the intercellular CO_2_ concentration in the treatment group decreased, while chlorophyll content showed no significant change. These changes in physiological indicators are fundamentally the result of gene regulation [40].

The signal transduction in plant response to drought stress involves a series of physiological changes, including signal perception, second messenger generation, and changes in gene expression. The ABA signaling pathway plays a crucial role in this process. Research has shown that *GmWRKY54* can influence soybean stomatal closure through ABA and Ca^2+^ signaling pathways [41]. Additionally, some transcription factor families are involved in signaling pathways outside the ABA signaling pathway. Studies have shown that *SRK2C/SnRK2.8* and *SRKF2F/SnRK2.7* from the SnRK2 family can directly respond to drought stress signals [42], causing stomatal closure through the phosphorylation of corresponding ion channels [43,44]. Furthermore, Ca^2+^ acts as an important second messenger in plant response to drought stress. When Ca^2+^ concentrations increase, the CIKP/SnRK3 family responds to Ca^2+^ signals by binding with CBL genes (Calcineurin B-Like), facilitating the transport of Na^+^ outside the cell through the phosphorylation of Na^+^/H^+^ antiporters on the cell membrane, thus maintaining ion balance inside and outside the cell [45].

Under water stress, leaf stomata close, leading to a decrease in photosynthetic rate due to insufficient CO_2_, a photosynthetic substrate [46]. Research has shown that in rice, the transcription factor *HYR* can enhance the expression of photosynthesis-related genes (*LHC2*, some photosystem II genes) under drought conditions, improving photosynthetic efficiency [47]. Additionally, studies have found that overexpression of the *CCA1/LHY* transcription factor in microalgae can increase carbon assimilation efficiency, leading to a 25% increase in carbohydrate storage in transgenic *Picochlorum celeri* [48].

The same stress condition can involve multiple transcription factors, and the same functional gene can be regulated by multiple transcription factors. Therefore, identifying more transcription factor regulatory genes that play crucial roles in enhancing crop stress resistance is significant for the genetic improvement of oats and promoting drought-resistant breeding of staple crops.

In this study, 344 differentially expressed genes were identified through transcriptome data analysis. KEGG and GO enrichment analyses revealed that seven differentially expressed genes in the photosynthetic carbon fixation pathway and three differentially expressed genes in the pentose phosphate pathway were enriched, with their gene expression levels significantly decreased. These ten genes are involved in regulating glucose-6-phosphate dehydrogenase (*A_satnudSFS4A01G001930* and *A_satnudSFS4D01G005410*), fructose-bisphosphate aldolase (*A_satnudSFS4D01G005932* and *A_satnudSFS4A01G001345*), phosphoglycerate kinase (*A_satnudSFS1A01G001130*, *A_satnudSFS1D01G003897*, and *A_satnudSFS1A01G004166*), and Rubisco (*A_satnudSFS6A01G005280* and *A_satnudSFS6A01G005281*). It was also found that the expression of genes regulating dicarboxylate transporter proteins related to Na^+^ ion transport showed a significant decrease. The expression of all regulatory genes of heavy-metal-associated isoprenylated plant proteins showed a significant decrease. The expression of genes regulating Annexins and chloride channel protein showed a significant increase.

The significant downregulation of Na^+^ ion transport-related genes and the significant upregulation of Ca^2+^ ion transport-related genes, combined with the decrease in stomatal conductance with increasing drought stress time, indirectly reflect the promotion of Na^+^ transport out of guard cells, facilitating stomatal closure [49]. Additionally, it can be speculated that the mechanism by which Ca^2+^ induces stomatal closure involves an increase in cytosolic Ca^2+^ concentration during drought stress in oats [50]. Concurrently, the expression of the *A_satnudSFS5C01G003567* gene shows a significant upregulation, activating Annexins and calcium-dependent protein kinases (CPKs). Phosphorylation by CPKs activates plasma membrane Cl^−^ ion channel proteins [51], which can be evidenced by the significant upregulation of *A_satnudSFS2C01G002908* and *A_satnudSFS2D01G006819* genes involved in the regulation of Cl^−^ ion channel proteins. Activation of Cl^−^ ion channel proteins leads to the efflux of anions (Cl^−^), and this efflux drives K^+^ through outward rectifying K^+^ (K^+^_out_) channels [52], reducing the turgor pressure of guard cells and thus causing stomatal closure. Moreover, the increase in Ca^2+^ concentration inhibits inward rectifying K^+^ (K^+^_in_) channels, which further prevents stomatal opening [53]. The reduced stomatal aperture leads to decreased stomatal conductance and transpiration rate.

The decrease in stomatal conductance leads to a drop in intercellular CO_2_ concentration. This reduction lowers the activity of five key enzymes that use CO_2_ as a substrate in the photosynthetic carbon assimilation pathway and the pentose phosphate pathway, including Rubisco, fructose-1,6-bisphosphate aldolase, and glucose-6-phosphate dehydrogenase. Consequently, photosynthesis is inhibited, resulting in a decrease in the net photosynthetic rate.

Gene co-expression analysis combined with promoter analysis revealed that 20 transcription factors, including *A_satnudSFS2C01G003971*, *A_satnudSFS6D01G001494*, *A_satnudSFS7C01G002678*, *A_satnudSFS2D01G000591*, and *A_satnudSFS7A01G002691*, have binding sites within the genes regulating 12 proteins. These transcription factors may be involved in regulating the aforementioned 17 key genes.

Genes related to photosynthesis play a crucial role in regulating photosynthesis in oats and in resisting abiotic stress. Current research has demonstrated that TERF transcription factors significantly impact the splicing of introns during chloroplast gene transcription [54]. In rice, the transcription factor *OsDREB1C* has been found to regulate the photosynthetic gene *RBCS3* [55]. This study also identified binding sites in the promoters of the glucose-6-phosphate 1-dehydrogenase gene for the *DREB1A*-type transcription factors *A_satnudSFS5A01G005245* and *A_satnudSFS5C01G004248*. Additionally, *MYB*-related transcription factors such as *A_satnudSFS6A01G001999*, *A_satnudSFS6C01G005376*, and *A_satnudSFS2C01G003971* were found to have binding sites in the promoters of photosynthetic carbon assimilation-related genes. However, there is limited research on how MYB transcription factors regulate plant photosynthesis. In Arabidopsis, *AtMYB28* and *AtMYB29* transcription factors have been shown to play a significant role in the specific expression of photosynthesis genes in mesophyll and bundle sheath cells [56]. Birch tree *MYB106* can bind to promoters of some photosynthesis-related genes to activate their transcription [57]. Additionally, *CDF3*-type transcription factors *A_satnudSFS1D01G000953* and *A_satnudSFS4A01G004380*, as well as *CDF2*-type transcription factor *A_satnudSFS4C01G000546*, were found to have binding sites in the promoters of photosynthetic carbon assimilation-related genes.

This study analyzed the dynamic changes in physiological indicators of oats under drought stress, identified transcription factors involved in regulating photosynthesis and influencing stomatal opening and closing, and analyzed the mechanism by which oat ion channel genes affect stomatal regulation. The results will provide a basis for future selection of high-quality drought-resistant genes from the oat genome, elucidating the biological functions of these genes and laying a foundation for the genetic breeding of oat.

## 5. Conclusions

Through the measurement of photosynthesis and analysis of leaf transcriptomes, this study identified 10 genes associated with plant photosynthetic carbon assimilation and 8 differentially expressed genes encoding metal ion and Cl ion transport proteins. These 10 genes encode six different proteins, including ribulose bisphosphate carboxylase (P00871 and Q40004), fructose bisphosphate aldolase (Q40677), phosphoglycerate kinase (P12782), glucose-6-phosphate 1-dehydrogenase (Q43839), and NADP-dependent glyceraldehyde-3-phosphate dehydrogenase (Q8LK61). Eight differentially expressed genes associated with metal ion and Cl ion transport were Dicarboxylate transporter 2 (Q8L7Z9), Tonoplast dicarboxylate transporter (Q8LG88), heavy-metal-associated isoprenylated plant protein (Q9C5D3), chloride channel protein (Q96282), and Annexin (Q94CK4). Through gene co-expression analysis combined with promoter region structural analysis, 11 transcription factors (from *MYB*, *AP2/ERF*, *C2C2-dof*) regulating key enzymes of photosynthetic carbon assimilation, 15 transcription factors (from *NAC*, *MYB*, *zf-HD*, etc.) regulating metal ion transport proteins, 5 transcription factors (from *MYB*, *bZIP*, *AP2/ERF*) regulating chloride channel protein, and 2 transcription factors (from *NAC*, *MYB*) regulating Annexin were identified.

## Figures and Tables

**Figure 1 plants-13-03317-f001:**
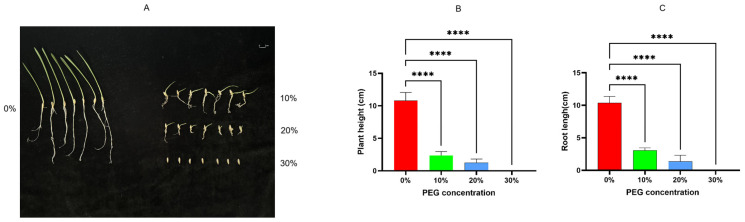
The inhibitory effects of different concentrations of PEG-6000 on the root and plant height of oats. (**A**) The germination status of oat seeds after being treated with PEG-6000 for 9 d. (**B**) Plant height of oat seedlings after treatment with four concentrations of PEG. (**C**) Root length of oat seedlings after treatment with four concentrations of PEG. Asterisks indicate statistical significance between the 0% treatment group and the 10%, 20%, and 30% treatment groups (****, *p* < 0.0001).

**Figure 2 plants-13-03317-f002:**
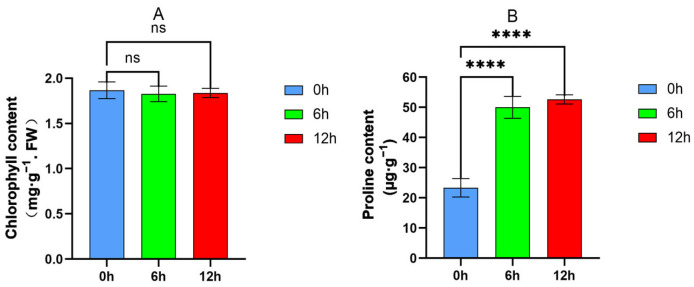
Chlorophyll and proline content in oat leaves after drought treatment 0 h, 6 h and 12 h. (**A**) Chlorophyll content of oat leaves. (**B**) Proline content of oat leaves. Asterisks indicate statistical significance between the 0 h treatment group and the 6 h and 12 h treatment groups (ns, *p* > 0.05; ****, *p* < 0.0001).

**Figure 3 plants-13-03317-f003:**
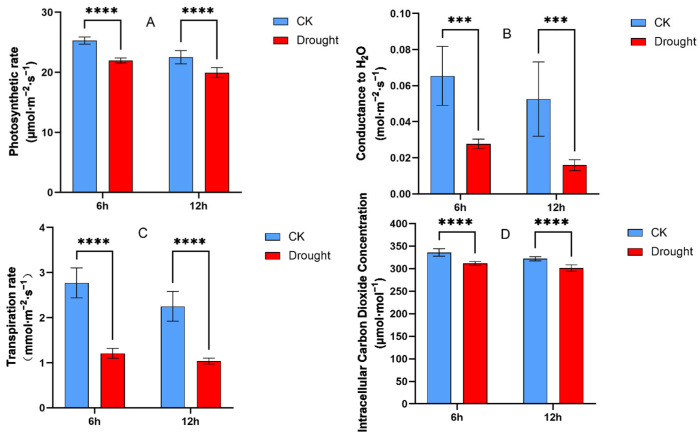
Photosynthetic parameters of oat leaves after drought treatment 0 h,6 h and 12 h. (**A**) Net photosynthetic rate, (**B**) stomatal conductance, (**C**) transpiration rate, (**D**) intracellular carbon dioxide concentration. Asterisks indicate statistical significance between the 6 h and 12 h treatment groups (***, *p* < 0.001; ****, *p* < 0.0001).

**Figure 4 plants-13-03317-f004:**
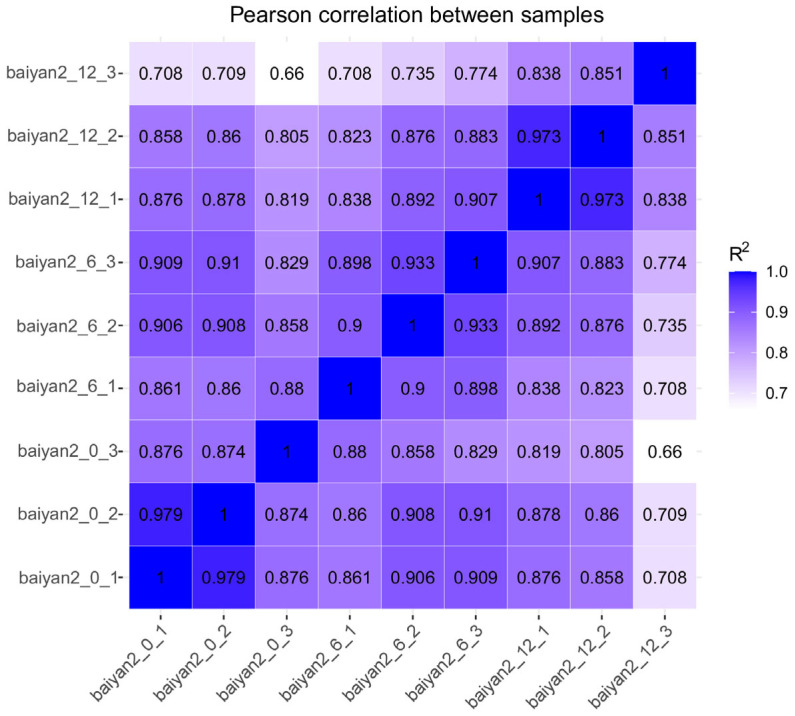
The Pearson correlation of gene expression levels in 12 oat leaf samples, with baiyan2_0_1, baiyan2_0_2, and baiyan2_0_3 representing the 0 h sampling group after drought treatment. baiyan2_6_1, baiyan2_6_2, and baiyan2_6_3 were the 6 h sampling group after drought treatment, and baiyan2_12_1, baiyan2_12_2, and baiyan2_12_3 were the 12 h sampling group after drought treatment.

**Figure 5 plants-13-03317-f005:**
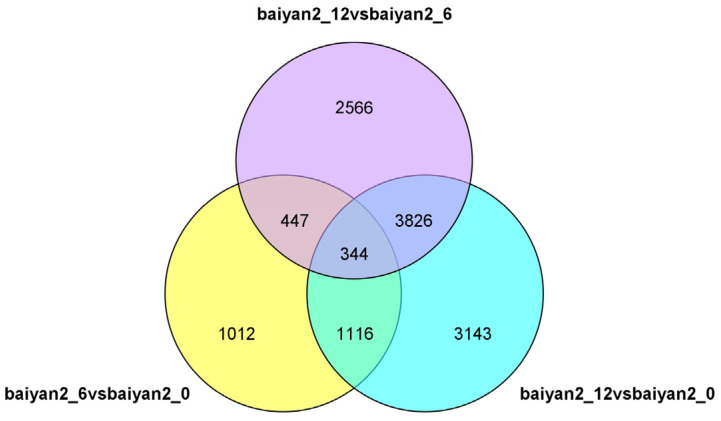
Venn diagram showing the overlap of differentially expressed genes among the three different groups. baiyan2_6vsbaiyan2_0 represents the differentially expressed genes in oat seedlings under drought stress at 6 h compared to 0 h. baiyan2_12vsbaiyan2_0 represents the differentially expressed genes in oat seedlings under drought stress at 6 h compared to 0 h. baiyan2_12vsbaiyan2_6 represents the differentially expressed genes in oat seedlings under drought stress at 6 h compared to 0 h.

**Figure 6 plants-13-03317-f006:**
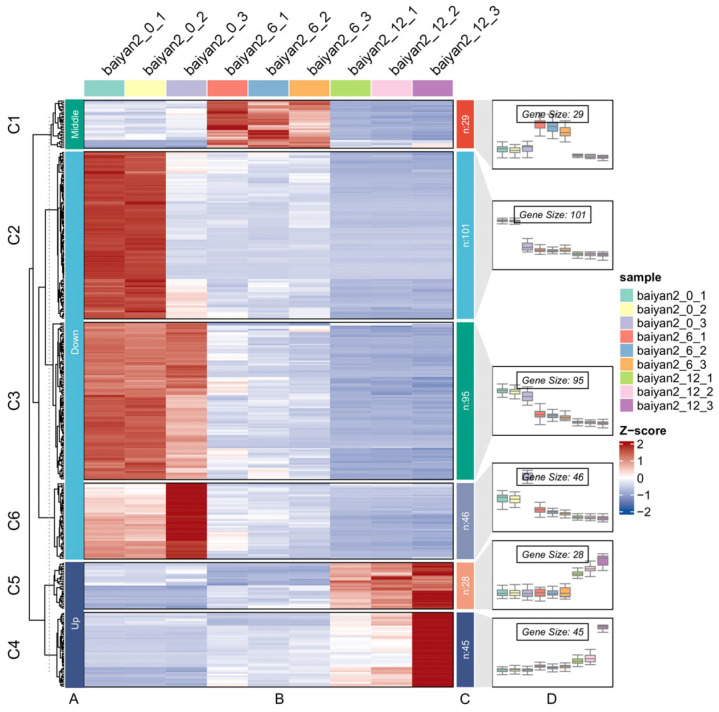
The expression patterns and clustering heatmap of the 344 DEGs. (**A**) Group of expression patterns. (**B**) Heatmap of the 344 DEGs. (**C**) Number of genes in each group. (**D**) Gene expression pattern of the sample. The Z-score represents the relative expression level of a gene, which is a normalized value; red indicates high expression, while blue indicates low expression.

**Figure 7 plants-13-03317-f007:**
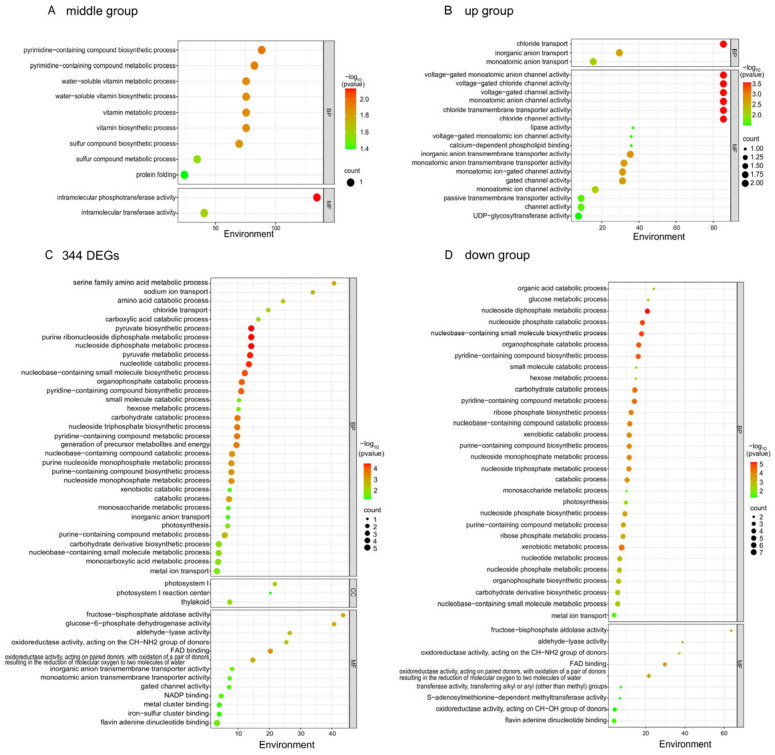
Gene ontology (GO) enrichment of differentially expressed genes (DEGs). (**A**) GO enrichment of upregulated genes in oat seedlings treated with PEG-6000 for 6 h (middle group), (**B**) GO enrichment of upregulated genes in oat seedlings treated with PEG-6000 for 12 h (up group), (**C**) GO enrichment of all DEGs (344 DEGs), (**D**) GO enrichment of upregulated genes in oat seedlings treated with PEG-6000 for 0 h (down group). The legend colors represents the −log10 (*p*-value) of the enrichment test; the size of the circle represents the number of genes.

**Figure 8 plants-13-03317-f008:**
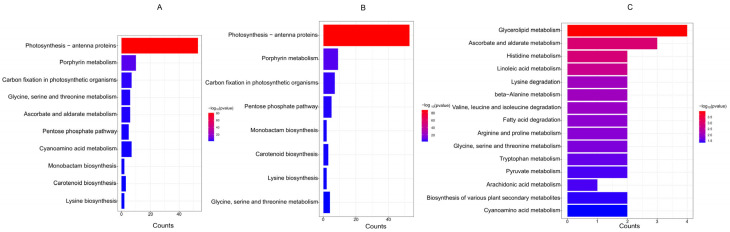
KEGG enrichment of differentially expressed genes. (**A**) KEGG enrichment of 344 DEGs, (**B**) KEGG enrichment of 242 downregulated genes (down group genes), (**C**) KEGG enrichment of 73 upregulated genes (up group genes). The legend colors represents the −log10 (*p*-value) of the enrichment test.

**Figure 9 plants-13-03317-f009:**
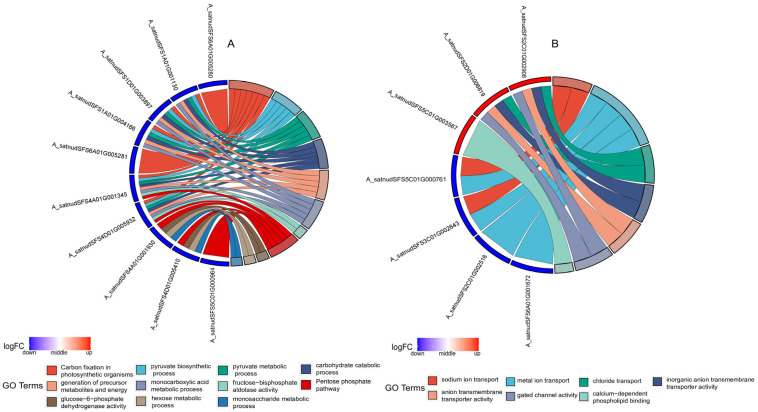
Chord diagram of differentially expressed genes associated with photosynthetic carbon fixation and ion transport and their corresponding GO terms. (**A**) Chord diagram of key genes in photosynthetic carbon fixation and their corresponding GO terms, (**B**) chord diagram of key genes in ion transport and their corresponding GO terms. Gene expression levels are represented by logFC, with logFC values of −1 for downregulated genes and 1 for upregulated genes.

**Figure 10 plants-13-03317-f010:**
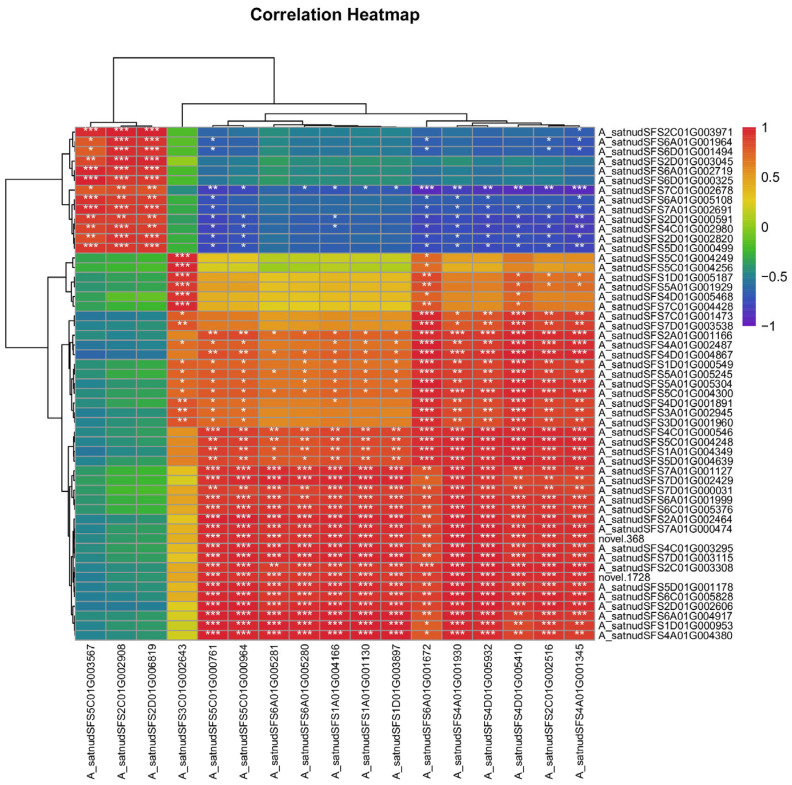
Correlation analysis between 17 key genes and 53 transcription factors. The numerical values in the small cells represent the correlation coefficients, and the colors represent the *p*-values of the correlation coefficient tests (*, *p* < 0.05; **, *p* < 0.01; ***, *p* < 0.001).

**Figure 11 plants-13-03317-f011:**
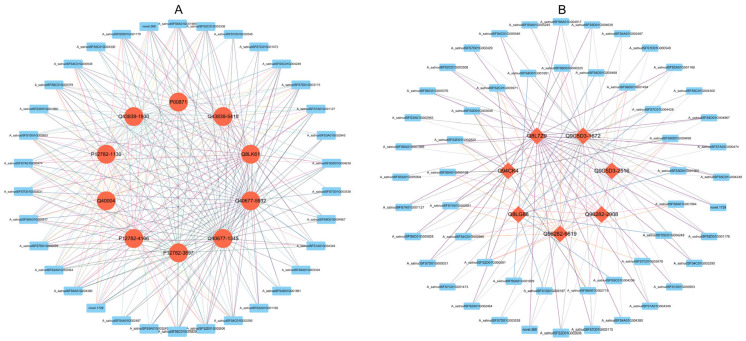
Transcription factor and gene regulatory network: the relationship between transcription factors and genes is based on gene co-expression analysis and transcription factor binding analysis in gene promoter regions. (**A**) Network diagram of key genes involved in photosynthetic carbon assimilation. (**B**) Network diagram of key genes in ion transport.

**Table 1 plants-13-03317-t001:** Protein Classification.

Gene Id	Protein Id	Protein Description
*A_satnudSFS6A01G005280*	P00871	Ribulose bisphosphate carboxylase small subunit
*A_satnudSFS1A01G001130*	P12782	Phosphoglycerate kinase
*A_satnudSFS1D01G003897*		
*A_satnudSFS1A01G004166*		
*A_satnudSFS6A01G005281*	Q40004	Ribulose bisphosphate carboxylase small subunit
*A_satnudSFS4A01G001345*	Q40677	Fructose-bisphosphate aldolase
*A_satnudSFS4D01G005932*		
*A_satnudSFS4A01G001930*	Q43839	Glucose-6-phosphate 1-dehydrogenase
*A_satnudSFS4D01G005410*		
*A_satnudSFS5C01G000964*	Q8LK61	NADP-dependent glyceraldehyde-3-phosphate dehydrogenase
*A_satnudSFS5C01G000761*	Q8L7Z9	Dicarboxylate transporter 2
*A_satnudSFS3C01G002643*	Q8LG88	Tonoplast dicarboxylate transporter
*A_satnudSFS2C01G002516*	Q9C5D3	Heavy-metal-associated isoprenylated plant protein
*A_satnudSFS6A01G001672*		
*A_satnudSFS2C01G002908*	Q96282	Chloride channel protein
*A_satnudSFS2D01G006819*		
*A_satnudSFS5C01G003567*	Q94CK4	Annexin

**Table 2 plants-13-03317-t002:** Types of transcription factors with binding sites in key gene promoters.

Gene Id	The Protein Id Corresponding to the Regulated Transcription Factors	Gene Name	Family
*A_satnudSFS3D01G001960*	Q8LG88, Q9C5D3, Q43839	*ZHD5*	*zf-HD*
*A_satnudSFS3A01G002945*	Q9C5D3, Q43839		
*A_satnudSFS7C01G002678*	Q96282	*VIP1*	*bZIP*
*A_satnudSFS6A01G001999*	Q9C5D3, P12782, Q43839, Q40677, Q8LK61	*RVE6*	*MYB->MYB-related*
*A_satnudSFS6C01G005376*	Q9C5D3, P12782, Q43839, Q40677, Q8LK61		
*A_satnudSFS2C01G003971*	Q94CK4, Q96282	*RVE1*	*MYB->MYB-related*
*A_satnudSFS4D01G001891*	Q8LG88	*ORR4*	*Others*
*A_satnudSFS2D01G002820*	Q94CK4	*NAC54*	*NAC*
*A_satnudSFS7A01G001127*	P12782, Q43839, Q40677, Q8LK61	*LHY*	*MYB->MYB-related*
*A_satnudSFS7D01G002429*	P12782, Q43839, Q40677, Q8LK61		
*A_satnudSFS6D01G001494*	Q96282		
*A_satnudSFS4D01G005468*	Q8LG88	*LBD37*	*LOB*
*A_satnudSFS2D01G000591*	Q96282	*ERF2*	*AP2/ERF->AP2/ERF-ERF*
*A_satnudSFS7A01G002691*	Q96282		
*A_satnudSFS5A01G005245*	Q43839	*DREB1A*	*AP2/ERF->AP2/ERF-ERF*
*A_satnudSFS5C01G004248*	Q8L7Z9, P12782, Q43839		
*A_satnudSFS1D01G000953*	P12782, Q40004, P00871, Q8LK61	*CDF3*	*C2C2->C2C2-Dof*
*A_satnudSFS4A01G004380*	P12782, Q40004, P00871, Q8LK61		
*A_satnudSFS4C01G000546*	Q8L7Z9, Q9C5D3, P12782, Q40004, P00871, Q40677, Q8LK61	*CDF2*	C2C2->C2C2-Dof

## Data Availability

Data will be made available on request.

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
