# Peer review of "Molecular Regulation of Photosynthetic Carbon Assimilation in Oat Leaves Under Drought Stress"

_plants, 2024, doi:10.3390/plants13233317_

Round 1
Reviewer 1 Report
Comments and Suggestions for Authors
General comments:
Please use the same font size throughout the manuscript.
Figure captions must be given in more detailed manner.
The conclusion lacks specificity; therefore, it is recommended to incorporate a more detailed analysis of the observed results.
The introduction is too general and does not align with the flow of the research; therefore, it should be rewritten in a more straightforward manner.
Specific comments:
Lines 134-142 Please add the reference for Proline determination, since it is the standard method for proline accumulation detection.
In results, subsection 3.1 the results are missing. Please provide data in table or figure. Add information regarding the percentage of germinated seeds under all analyzed treatments.
Figure 1 caption should be paraphrased. I’ll suggest to the authors to use “drought” stress instead of “PEG”, in addition add the species name, since “Baiyan 2” is not completely clear. Also, add information regarding the statistical significance levels.
Figure 2. As in figure 1, add the significance level.
Figure 4. caption require more detailed explanation. Please add information regarding Venn diagram
Lines 283-285 add genes names.
Line 420 “disaster” is redundant, please delete
Sincerely,
The reviewer
Reviewer 2 Report
Comments and Suggestions for Authors
The study of Yiqun Xu and co-authors entitled “Molecular Regulation of Photosynthetic Carbon Assimilation in Oat Leaves under Drought Stress” is dedicated to a relevant topic. Identifying drought-resistant genes in crops, including oats, is crucial for genetic breeding. In general, the study is well-designed, and the methods used are adequate and modern. The results have both fundamental and practical value.
However, some correction is needed to improve the quality of the paper presentation:
Abstract. Add a final sentence describing the importance of obtained results both in fundamental and practical points of view.
Introduction. Line 91 - Rephrase the sentence “In this study, Baiyan 2 was used as the experimental material”. Clarify information about Baiyan 2 – it this an oat cultivar? Is it а drought-tolerant cultivar?
Material and methods. Line 99 – provide a Latin name for oat, as well as information about the sensitivity of this cultivar to drought (drought-tolerant or drought-sensitive?).
Line 106 – Did you used PEG-6000? Clarify, please.
Lines 125-133, 134-142, 143-150, 151-164, 165-184. Provide references to the used methods.
Add subsection Statistical analysis.
Results. Figs. 2, 6, 7, 8, 9, 10. It is difficult to read the information from graphics, the fonts of the text should be enlarged.
Fig. 3 should be enlarged.
Fig. 4. “Venn of DEGs“ Give more detailed description.
Recommendation. Adding a separate Conclusion section would make it easier for readers to perceive information about the specific results obtained and their fundamental and practical significance.
The entire text of the manuscript must be double-checked for spelling and stylistic errors, as well as and for compliance with the requirements of the journal (https://www.mdpi.com/journal/plants/instructions).
Best wishes,
Reviewer
Reviewer 3 Report
Comments and Suggestions for Authors
This paper reported the effects of short-term PEG-simulated drought stress imposed on oat seedlings on chlorophyll content, proline, net photosynthesis rate and transcriptome changes. They identified differentially expressed proteins related to carbon fixation, pentose phosphate pathway and ion transport, and a number of transcription factors potentially regulating these genes. Although the physiological and molecular data discovered are not novel, the paper is still worth being published considering relatively rare molecular data from oats in response to drought stress. There are some points in the manuscript that should be improved as follows:
(1) Results: - The legends of Figures 3 to 9 should be more clearly written. Figure legends should be self-explanatory. Also explain the meaning of asterisks in Figures 1 and 2. Figure 6 legend – please indicate more clearly ‘the up group’ / ‘the down group’.
(2) Materials and methods – 1) please indicate the reference for analysis of chlorophyll and proline. 2) please indicate the light intensity and time of measurement for leaf gas exchange.
(3) Discussion -1) while discussing regulation of ion transport (Lines 479 – 487), please try to relate the previous knowledge (please also add references) with your current data. 2) The same for discussing transcription factors relating to photosynthesis. 3) Please rewrite paragraph (Lines 471 – 477), especially the sentence ‘When involved in metabolic pathways, glucose-6-phosphate dehydrogenase, …….. produce CO2”?? Also, the sentence ‘This demonstrated that the impact of drought stress on photosynthesis primarily occurs in the photosystem” ?? The authors did not measure fluorescence parameters or any parameters occurring in the light reactions, how can they claim that the impact of stress on photosynthesis occurs in the ‘photosystem’.
